# Physicochemical and Sensorial Profile of Meat from Two Rabbit Breeds with Application of High-Intensity Ultrasound

**DOI:** 10.3390/foods14061059

**Published:** 2025-03-20

**Authors:** Isaac Jhonatan Vargas-Sáenz, Iván Adrián García-Galicia, Alma Delia Alarcón-Rojo, Felipe Alonso Rodríguez-Almeida, Martha María Arévalos-Sánchez, Luis Manuel Carrillo-López, Teresita de Jesús Hijuitl-Valeriano, Mariana Huerta-Jiménez

**Affiliations:** 1Facultad de Zootecnia y Ecología, Universidad Autónoma de Chihuahua, Periférico Francisco R. Almada km 1, Chihuahua 31453, Chihuahua, Mexico; jhonatan.vargas@uaq.mx (I.J.V.-S.); aalarcon@uach.mx (A.D.A.-R.); frodrigu@uach.mx (F.A.R.-A.); marevalos@uach.mx (M.M.A.-S.); 2Centro de Enseñanza, Investigación y Extensión en Ganadería Tropical, km. 5.5 Carr. Fed. Martínez de la Torre-Tlapacoyan, Mpio. de Tlapacoyan 93650, Veracruz, Mexico; 3Consejo Nacional de Humanidades, Ciencia y Tecnología, Av. Insurgentes sur 1582, Crédito Constructor, Demarcación Territorial Benito Juárez, Ciudad de México 03940, Mexico; lmcarrillo@uach.mx; 4Facultad de Ciencias Naturales, Universidad Autónoma de Querétaro, Av. de las Ciencias s/n, Juriquilla, Santa Rosa Jáuregui 76230, Querétaro, Mexico; teresita.hijuitl@uaq.mx

**Keywords:** cavitation, quality, muscle, tenderness, *Oryctolagus cuniculus*, rabbit breed

## Abstract

The application high-intensity ultrasound (HIU) in rabbit meat has shown promising results for improving its palatability and commercial value. This study aimed to evaluate the effect of HIU application in meat from two rabbit breeds (Flemish Giant, FG and Azteca Negro, AN) on its physicochemical parameters (PQs) and sensory profile (SEN). Five carcasses of each breed were frozen and dorsally cut into half carcasses. HIU was applied (20 min, 50 kHz, and 200 W) to one randomly selected half carcass of each rabbit. PQs evaluated were pH, color (*L**, *a**, and *b**), chroma (C*), hue angle (HUE), water holding capacity (WHC), collagen content, and shear force (SF). Color, odor, flavor, and texture were evaluated for the SEN. Statistical analysis was performed using ANOVA, adjusting a general mixed model with the fixed effects of treatment, breed, and their interaction. Significant interaction differences (*p* < 0.05) of breed and HIU were observed in the collagen content. Breed had an effect (*p* < 0.05) on pH, *a**, *b**, C*, and on the descriptors of the sensory attribute color. HIU had an effect (*p* < 0.05) on *L**, *a**, HUE, C*, and SF, as well as on sensory descriptors like texture and color. HIU increases the physicochemical and sensory perception of the tenderness of the meat of both breeds.

## 1. Introduction

Rabbit meat production with a focus on obtaining meat has undergone constant development. The development of breeds with both genotypic and phenotypic specialized traits for meat production has gone hand in hand with genetic selection models to drive these traits further. The quality of carcass and meat is closely related to the breed of the animal [1], even in breeds of different sizes [2]. The chemical composition of meat is also different among breeds, even if they are of the same size [3,4].

Some authors have reported that breed determines the differences in carcass and meat quality in rabbits [1,5]. For instance, the Flemish Giant breed is one of the most popular giant breeds in the world [6]. However, despite producing carcasses of up to 4 kg, this breed is of little commercial importance in countries such as Mexico, since its meat is commonly thought to be tougher compared to a medium-sized breed (Black Azteca, of Mexican origin) [7].

The application of high-intensity ultrasound (HIU) causes changes in physical and chemical properties of meat and its products. Hence, researchers have shown a high degree of interest in this technology because it can represent an alternative to chemical or thermal processes [8]. The positive effect of ultrasound in meat has been associated with the complex phenomena of acoustic cavitation, which refers to the generation and evolution of microbubbles in a liquid medium [9]. The beneficial effects of ultrasonication in foods of animal origin are promoting mass transfer, the activation or inhibition of enzymes, reduction in microbial load, higher emulsification, crystallization, homogenization, cell breakdown, and improvement in organoleptic characteristics such as color and tenderness [8].

Particularly, the application of HIU has been reported to be of high potential to improve the characteristics of rabbit meat. The most important identified effects are the reduction in water holding capacity (WHC), decrease in muscle pH, reduction in SF, changes in color parameters, and decrease in marinating times [10,11,12]. Therefore, the objective of this study was to observe the effect of HIU on the physicochemical and sensory properties of rabbit meat from two breeds of different sizes.

## 2. Materials and Methods

Two breeds of male rabbits were tested in this study, Azteca Negro (AN, n = 5, 90 d old) and Flemish Giant (FG, n = 5, 150 d old). The rabbits of both breeds were obtained from a commercial farm under the same production and feeding conditions. The slaughter was carried out according to the ‘Bioethical code and Regulations of animal welfare’ of the Facultad de Zootecnia y Ecología, Universidad Autónoma de Chihuahua (UACH 2007, official number P/302/2017). Those regulations are based on the Official Mexican Regulations for slaughtering domestic and wild animals (NOM-033-SAG/ZOO-2014): “Methods for slaughter of domestic and wild animals” through desensitization by neck breaking [13]. The slaughter was carried out by bleeding through the jugular cutting. The process of converting muscle to meat was carried out for 24 h at 4 °C. The carcasses were identified and refrigerated for 24 h post-mortem and then fast frozen (−20 °C) to be transported (0 °C for 24 h). The carcasses were cut longitudinally along the spine, obtaining half rabbit carcasses which were weighed (AN = 628.61 ± 14.5 and FG = 1192.27 ± 8.24 g/each half carcass) and later vacuum-packed individually to be randomly distributed into two treatments as replications (right sides for ultrasonication and left sides for control). After thawing the half carcasses at room temperature (~20 °C) for 2 h, HIU was applied for 20 min (10 min/each side of the half carcass) at 50 kHz and 200 W (Hielscher^®^ UP400St, Teltow, Germany). The ultrasonication equipment was manipulated according to the instructions of the manufacturer. The diffusion medium for the HIU and control was deionized water (3 to 5 °C). All carcass halves were kept in the same location of the supporting platform during treatment, seeking to locate the probe hit on the center of the carcass. The temperature was monitored with a thermocouple connected to a portable thermometer (Thermo Fisher^®^, Waltham, MA, USA), while the bath temperature was kept constant with a portable immersion cooler (Julabo^®^ FT200, Seelbach, Germany). The HIU parameters were chosen based on previous studies finding positive effects when comparing time, frequency, and power on the meat of rabbit and other animal species [10,11]. The control treatment consisted of half carcasses being kept under the same conditions for 20 min. 

### 2.1. Physicochemical Analysis

The variables, which included pH, luminosity (*L**), red tendency (*a**), yellow tendency (*b**), color intensity (chroma, C*), hue angle (HUE), water holding capacity (WHC), determination of collagen, and shear force (SF), were evaluated for the PQ analysis. The pH was recorded according to the methodology of Honikel [14] using a pH meter (Hanna^®^ model HI99161N, Cluj-Napoca, Romania). The measurements were taken in triplicate from the external m. *Biceps femoris*. The color parameters *L**, *a**, *b**, and C^*^ were obtained with a colorimeter (Konica Minolta, CR 400, Indianapolis, IN, USA. Aperture, 8 mm. Illuminant C. Standard observer, C: Y = 94.2, x = 0.3130 and y = 0.3190) calibrated with a white calibration plate, according to the instructions of the manufacturer. The measurement was carried out under the CIE (Commission Internationale Pour I’Eclarige) reference system [15]. The HUE was calculated according to the equation: h = arctan (*b**/*a**). The WHC was determined in triplicate according to the compression method proposed by Tsai and Ockerman [16], from the external m. *Biceps femoris*. The results were expressed as the percentage of exudate released. The shear force measurement was carried out following the methodology set by the AMSA [17]. The m. *Longissimus dorsi* samples were cooked individually inside food-grade polyethylene bags in a water bath (ISOTEMP 215), until 71 + 1 °C was reached in the geometric center of the sample. Six cylinders of 1 cm diameter were obtained per animal. These were cut perpendicularly to the direction of the fibers using a Warner-Bratzler “V” shape blade (60° triangular opening) at a speed of 1 mm/s and 20 mm. The maximum peak force was recorded during the test and expressed in Newtons (N) with a TA-XT plus texture analyzer (Stable Micro Systems Ltd., Surrey, UK). The determination of the hydroxyproline content was determined by the ISO 3496 methodology [18]. A spectrophotometer (Thermo Spectronic, 4001/4, Waltham, MA, USA) was used to measure the absorbance of the oxidation reaction at 558 nm, and the result was expressed as the content of hydroxyproline per gram of sample. Subsequently, a conversion factor of 7.25 was used to calculate the collagen content [19].

### 2.2. Sensory Profile Analysis

The sensory evaluation was carried out with a semi-trained panel (eleven panelists) performing the descriptive quantitative analysis technique described by Peña-Gonzalez et al. [20] and the use of ISO 4121 intensity scales [21]. The group of panelists is part of our animal science department at UACH; they are proficiently trained in meat and meat products’ sensory evaluation. They were all informed of the studies involved, including the present one, and they give their informed consent, ensuring their voluntary participation and understanding of the experimental objectives, in agreement with the “Políticas del Comité de Bioética de la UACH, 2022”. Briefly, the panelists were selected and trained under the descriptive quantitative analysis technique described by Meligaard et al. [22]. The panelists were volunteers, of both sexes (23 to 67 y old), all of them consumers of rabbit meat. They were selected based on the results of tests such as the basic color identification test, shape identification, basic flavor identification, basic odor identification, hardness, texture, and a test profile. All of them provided correct answers higher than 80% in all of the tests. The training also included familiarization with relevant descriptive terms, selection and quantification of sensory characteristics of cooked meat, as well as the use of ISO 4121 [21] intensity scales. The specific training for this study lasted around 13 h. It was carried out with specific tests of rabbit meat profile: muscle fiber tenderness ranking, connective tissue ranking, juiciness ranking, muscle fiber tenderness intensity measurement, juiciness intensity measurement, and connective tissue intensity measurement. The selected descriptors are described in Table 1. The intensity of the descriptor for each attribute was evaluated on a 10 cm linear scale with two points (0 = less intense and 10 = higher intensity). The muscle samples (*Supraspinatus*, *Infraspinatus*, and *Triceps brachii*) were taken from the scapular area and boiled individually inside food-grade polyethylene bags at a temperature of 85 °C in a water bath (Fisher Scientific, ISOTEMP 215, Schwerte, Germany) until 71 ± 1 °C was reached in the geometric center of the sample [17]. Temperature was monitored with a thermocouple placed at the geometric point of the sample. The sensory analysis was performed for both the sonicated (HIU) and control samples (C) in one session. For this evaluation, the entire scapular muscle portion of the rabbit was used. Samples were cut into eleven equal pieces of approximately 30 g and kept at 35 °C until sensory analysis (≤30 min). Testing was conducted in the Sensory Analysis Laboratory at room temperature (20 °C), 80% relative humidity, and white light. Panelists were instructed to cleanse their palates between samples using water and unsalted crackers, as well as the olfactory system with coffee support. The panelists received a randomly selected 30 g sample for each treatment group, identified by a three-digit code.

### 2.3. Statistical Analysis

The data were analyzed using an analysis of variance by fitting a mixed general linear model with the fixed effects of treatment, breed, and their interaction, as well as the random effect of the animal within the breed. For this purpose, the PROC MIXED procedure of the SAS 9.1.3 statistical program (SAS Inst. Inc., Cary, NC, USA) was used, and means were contrasted with Tukey’s test (*p* < 0.05).

## 3. Results

### 3.1. Physicochemical Variables

The effect of the breed on the variables WHC, SF, *L**, HUE, and collagen was not significant (*p* > 0.05). However, significant differences (*p* < 0.05) were observed in terms of pH, *a**, and *b** (Table 2).

The application of HIU was not statistically significant (*p* > 0.05) on pH, WHC, *b**, and HUE variables. But, for the variables *L**, *a**, chroma, WHC, SF, and collagen, it was significant (*p* < 0.05) (Table 2). An interaction (*p* < 0.05) was observed in the collagen variable of the effect of breed and HIU treatment (Table 2).

### 3.2. Sensory Profile

The effect of the breed on the sensory profile showed a significant effect on pink color, while the treatment had an effect on pale color, and there was an interaction only on shiny brown (Table 3).

In the texture attribute descriptors (Table 4), the effect of the HIU treatment generated a significant increase (*p* < 0.05) in the perception of softness.

## 4. Discussion

The number of animals can be considered to be limited in the present study. This limitation was mainly due to the availability of individuals with the characteristics previously defined in the objectives of this research. A larger sample is recommended for future studies. However, the main results of this study regarding the effect of HIU on meat coincide with results previously obtained by our research group in rabbit meat [10,11]. Hence, the effect of breed and the interaction of the main factors found in the present study resulted in being remarkable and relevant.

In the present study, only the breed factor in the pH variable exerted significant differences (*p* < 0.05) between the AN and FG breeds (*p* = 0.008); these differences between breeds were also observed in the work presented by Dalle Zote et al. [5]. Their results are similar to ours when comparing a medium-sized breed (California pH = 5.92) with a giant-sized breed (Gray Flemish Giant pH = 6.14). They concluded that the differences in pH were due to the breed (*p* < 0.05). In addition, their values, such as those observed in this experiment, fall within the standard for rabbit meat (pH = 5.5–6.1).

Lightness was not different between the two breeds in the present study. However, there was an increase in lightness in both breeds (*p* = 0.022). Some authors have mentioned that an increase in lightness in ultrasonicated meat usually represents a higher amount of surface water in the tissue [23,24]. In ultrasonicated meat, this is due to the release of exudate by opening up cellular microchannels and pressures and decompressions of the ultrasound energy. Lightness is a key factor for consumer acceptance because it improves the appearance of meat by reflecting light better [23,25]. The values of the chromatic coordinates *a** (redness) and *b** (yellowness) were different between both breeds. The AN breed had higher *a** values (7.56) than the FG breed (4.49). This difference between breeds coincides with that mentioned by Dalle Zotte et al. [26], who compared color between medium- and giant-sized breeds. They determined that color differences are related to the quantities of type I muscle fibers, since giant breeds present lower quantities in their muscles in comparison to medium-size breeds. The type I fibers of muscle are linked to a higher pigmentation of the meat since they have a higher quantity of myoglobin compared to type II.

HIU application generated decreases in *a** in both breeds (*p* = 0.004). This effect may be related to the denaturation of the muscle color pigments (myoglobin and hemoglobin) generated by the acoustic cavitation by the increase in temperature in the medium-sized breed or by the lixiviation of the pigments by the rupture of structures in the tissue [8]. The *b** value was different between breeds (*p* = 0.009): AN had higher value (5.88) in comparison to FG (2.83). The color of the meat depends on the levels of oxidation reduction in myoglobin. The *b** values increase at higher rates of the oxido-reduction process [27,28]. It is important to note that the HIU application did not generate changes in the muscle that altered the *b** value. This is positive since Reyes-Villagrana et al. [10] pointed out that an increase in *b** has an undesirable effect on the quality of rabbit meat. Nasr et al. [29] reported that *a** and *b** in meat can be variable among breeds (New Zealand, Rex, and California). Likewise, Wang et al. [30] reported variable values among breeds (Hyla, Champagne, and Black Tianfu). In the present study, differences in HUE were found by HIU’s effect (*p* = 0.050). In this regard, the increase in the HUE angle is related to orange tones in the meat. This effect by HIU on rabbit meat has also been reported by Carrillo-Lopez et al. [11], which does not represent a negative effect on consumers acceptance. The color chroma (C*) was different between the two breeds (*p* = 0.006), while HIU application decreased it (*p* = 0.008) in both breeds, as more opaque tones were obtained compared to the control. Changes in color HUE and C* in sonicated meat are related to the instability of hemopigments when ultrasonication is applied [8]. Under these conditions, water migrates out of the cells due to the destruction of the muscle structure by HIU, resulting in the alteration of protein conformations and increased light scattering in the meat [11].

Regarding the color descriptors, both factors, breed and HIU application, had a significant interaction in the descriptor shiny brown (Table 3). Alarcon-Rojo et al. [8] mentioned that HIU generates oxidative changes in the hemopigments of the meat which, when subjected to cooking temperatures (Maillard reaction), generates changes in the appearance of the meat. In the case of this experiment, the perception of the shiny brown descriptor increased significantly for the AN breed (*p* < 0.05), but on the contrary, the perception of this descriptor decreased for the FG breed (*p* < 0.05). The factor of treatment with HIU generated significant increases (*p* < 0.05) in the perception of the pale descriptor in both breeds. In the breed factor, significant differences were observed in the perception of pink, with the FG breed having the highest perception (6.4 points). This is related to what Dalle Zotte et al. [26] mentioned about the differences among breeds in the amount of type I fibers (red fibers) in rabbit meat. Giant breeds have a higher amount of these fibers; the amount of myoglobin is much higher, so the pink color in cooked meat was much more noticeable compared to the AN breed (3.97 points). Meat color results from complex interactions among the chemical composition (hemopigments), the metabolic properties of the muscle, and the post-mortem biochemical changes (oxidation of hemopigments), leading to its conversion into meat [31]. Hence, the breed of the animal defining its behavior, feeding, exercise, size, and other characteristics may result in being important for the color of their meat. Fadare and Arogbo [3] did not observe significant differences (*p* > 0.05) in the sensory perception of color when comparing the breeds New Zealand, Black Havana, Palomino, and California. Daszkiewicz et al. [2] mentioned that the main factors impacting the color of rabbit meat are diet, genetics, and animal welfare at the time of slaughter.

The WHC values were not different between breeds and HUI treatments. Although reductions in WHC percentage were observed, they were not significantly different (*p* > 0.05. Table 2). This is in contrast to that reported by other authors [10,11] who found significant reductions (*p* < 0.05) in WHC due to the effect of HIU treatment. Alarcon-Rojo et al. [8] pointed out that this effect on WHC could be a consequence of acoustic cavitation which, by exerting pressures and decompressions, generates the opening of microchannels in the cell wall, through which there is a constant flow of water. The values obtained in the AN breed (%WHC = 67.57 ± 1.59) in this experiment are similar to those observed by Lee et al. [32] in a commercial breed (medium-sized), where they reported WHC percentages of 64.73 (*p* < 0.05). On the other hand, Nasr et al. [29] reported in three breeds (New Zealand, Rex, and California) percentages between 53.3 and 55.8%, which are lower than the data obtained in this experiment. The FG breed presented average values of 72.03% WHC, which are similar to the data reported by Margatama et al. [33] in the Apus Bambu breed, which is a breed developed in Indonesia as a result of crossing the New Zealand × Flemish Giant breeds, which showed WHC values of 72.38%. This percentage level is positive since Warner et al. [34] mentioned that a high WHC percentage directly influences the increase in the perception of tenderness and juiciness.

The breed factor did not generate a significant difference (*p* > 0.05) for the shear force variable in the AN (10.76 ± 0.32 N) and FG (11.24 ± 0.32 N). These results contrast with Grepe [35], who mentioned that the meat of Flemish Giant rabbits has little business relevance since it is less tender compared to that of other breeds. The slaughter age of the Flemish Giant breed may be relevant since Daszkiewicz and Gugołek [2] mentioned that at 91 days, there were significant differences (*p* < 0.05) in the SF variable when comparing Flemish Giant (22.54 N) and a medium-sized breed such as California (14.72 N) meat. Likewise, this last value is higher compared to that obtained for the AN breed (10.76 ± 0.32 N). On the other hand, Margatama et al. [33] reported SF values for the Apus Banbu breed (New Zealand × Flemish Giant) of 36.09 N, while for the New Zealand breed, 30.98 N was reported. Reyes-Villagrana et al. [10] reported 13.04 N and Cantarero-Aparicio et al. [36] reported 29.9 N in the same breed. All the studies used the Warner Blatzer type blade technique. The effect of treatment with HIU generated significant decreases (*p* = 0.0001) in both breeds (AN = 8.51 N and GF = 8.3 N); this effect is similar to that reported by Reyes-Villagrana et al. [10], who found significant decreases due to the effect of treatment with HIU (*p* < 0.05). In their control treatment, they obtained values of 13.04 and a reduction of 11.3 N in the ultrasonicated *Longissimus dorsi* muscle.

Carrillo-Lopez et al. [11] reported significant decreases because of treatment with HIU for 20 min (*p* < 0.05). They reported initial values of 11.45 N in the control treatment, with reductions to 7.58 N in the HIU treatment. The application of HIU generates the creation of microbubbles producing pressures and decompressions, which generates a double effect of the breaking and denaturation of collagen macromolecules and the rupture of muscle tissue [37]. In addition, cavitation can generate the migration of minerals and substances such as enzymes contained in lysosomes that accelerate the proteolysis process of the structural proteins in the muscle cells [8,37].

Collagen concentration in meat resulted in a significant interaction (*p* < 0.05) of HIU treatment and breed (*p* = 0.0001). There was a decrease in the collagen content of the Flemish Giant breed with the application of HIU (from 4.26 mg/g to 3.16 mg/g) compared to the AN breed. The collagen content in FG can be compared with that reported by Pascual et al. [38], who measured collagen in the *Longissimus dorsi* of hybrid breeds using the hydroxyproline content determination technique (6.8 mg/g). On the other hand, Ariño et al. [39] also reported 6.8 mg/g in meat from 68-day-old rabbits. However, in the case of this experiment, no differences were perceived between the breeds. Blasco et al. [40] mentioned that the amount of collagen in rabbit meat is a determining factor for the tenderness of the meat since this species has genetically low amounts of collagen compared to other species for consumption. An advantage of the texture in rabbit meat compared to beef is the high solubility of collagen proteins, which allows it to increase tenderness when subjected to high temperatures [41,42]. HIU has been shown to be effective in reducing the amount of collagen in meat [43]. The effect of acoustic cavitation denatures and breaks down collagen macromolecules [44]. Caraveo-Suarez et al. [45] mentioned that the amount of collagen in HIU treatment applied for 20 min decreases over the days of storage (7–14 d) in beef. HUI generates microbubbles due to pressure and decompression in a liquid media [9]. This energy can generate a double effect in which there is the breaking and denaturation of collagen macromolecules and rupture of muscle tissue. All these changes in the tissue can promote the migration of minerals and substances such as enzymes contained in lysosomes, which accelerate the proteolysis process of the structural proteins of the muscle. Hence, the effect of HIU on the reduction in collagen and shear force of meat can be indirect. Additionally, from the results of the SEN analysis, the samples treated with HUI had significant increases in the perception of tenderness in both breeds, which could be due to the decrease in SF.

Regarding the descriptors of texture, only the HIU application generated significant differences in hardness and softness, generating a decrease in the perception of the hardness descriptor and an increase in the perception of the softness descriptor in both breeds. The perception of softness has a direct relationship with the values observed in terms of shear force [8].

Some authors have mentioned that HIU application increases the perception of the tenderness of meat by the effect of acoustic cavitation in the tissue, besides a decrease in the content of structural proteins such as collagen [8,37]. It is important to note that the observed perception of softness in the rabbit meat controls is quite high, just as the perception of hardness in the controls is quite low. This was similar to what was mentioned by Nasr et al. [29], who compared the sensory tenderness of rabbit meat from different breeds. The average values obtained for the New Zealand, Rex, and California breeds range from 4.1 to 4.3 points (on a 5-point scale). In this experiment, no differences (*p* > 0.05) were observed between the breeds, which coincides with what was observed in the comparison between the AN and FG breeds. Daszkiewicz et al. [2] reported not observing differences between the FG and California breeds in the perception of juiciness, in addition to observing average values for the same characteristic (FG = 3.0 points and California = 3.1, on a 5-point scale). Similar average values were also perceived in the present experiment (AN = 5.01 points and FG = 4.48, on a 10-point scale).

## 5. Conclusions

Breed size did not influence the physicochemical variables linked to the perception of meat tenderness (collagen, WHC, and SF). This same result was observed in the perception of the softness and hardness of the breeds analyzed. The meat of a giant-sized breed (FG) at 150 days of production presents levels equivalent to those of a medium-sized commercial breed of Mexican origin.

On the contrary, breed size presented differences in the measurement of the color profile. The sensory perception of the color profile also reflected this difference between the breeds studied in this experiment. The HIU generated increases in the measurement and perception of the softness of rabbit meat in both breeds. The amount of collagen and its decrease through the HIU treatment are related to these increases in the measurement and perception of softness. The meat of the FG breed presented greater results in the increase in the perception of softness compared to that of the medium-sized AN breed. HIU treatment causes changes in the color profile; however, these changes do not affect the quality of the product image. HIU treatment increases water loss in rabbit meat of both breeds without affecting the perception of juiciness.

## Figures and Tables

**Table 1 foods-14-01059-t001:** Description of the attributes of rabbit meat by the semi-trained panel.

Attributes	Descriptors
Color	Pale (P), pearly (p), shiny brown (SB), pink (PK), and whitish (W).
Odor	Raw meat (RM), metallic (CM), roasted meat (RT), cooked fresh meat (CFM), and boiled meat (BM).
Flavor	Metallic (M), jerky meat (JM), cooked fresh meat (CFM), chicken meat (CM), and greasy (G).
Texture	Soft (S), elastic (E), hard (H), fibrous (F), and juicy texture (J).

**Table 2 foods-14-01059-t002:** Physicochemical parameters of rabbit meat from two different breeds (Azteca Negro, AN or Flemish Giant, FG) under application of high-intensity ultrasound treatment (means ± E.E.).

Variable	Azteca Negro		Flemish Giant		*p*		
	Control	HIU	Control	HIU	B	TRAT	B × TRAT
pH	5.84 ± 0.03	5.76 ± 0.03	5.57 ± 0.03	5.63 ± 0.03	0.0080	NS	NS
WHC (%)	67.57 ± 1.59	65.09 ± 1.59	72.03 ± 1.59	67.16 ± 1.59	NS	NS	NS
SF (N)	10.76 ± 0.32	8.51 ± 0.32	11.24 ± 0.32	8.30 ± 0.32	NS	0.0001	NS
*L**	53.39 ± 0.74	55.76 ± 0.74	52.99 ± 0.74	56.55 ± 0.74	NS	0.0224	NS
*a**	7.56 ± 0.50	3.93 ± 0.50	4.49 ± 0.50	2.36 ± 0.05	0.0112	0.0037	NS
*b**	5.88 ± 0.39	3.93 ± 0.39	2.83 ± 0.39	3.11 ± 0.39	0.0085	NS	NS
HUE	41.58 ± 3.68	44.59 ± 3.68	43.62 ± 3.68	48.6 ± 3.68	NS	0.0503	NS
Croma	13.45 ± 0.55	7.86 ± 0.55	8.39 ± 0.55	4.02 ± 0.55	0.0058	0.0081	NS
Collagen (mg/g)	3.67 ^b^ ± 0.094	3.55 ^b^ ± 0.094	4.26 ^a^ ± 0.094	3.19 ^c^ ± 0.094	NS	NS	0.0001

NS = no significant statistical differences were observed (*p* > 0.05), HIU = treatment high-intensity ultrasound, TRAT = effect of application of HIU, B = effect of breed, B × TRAT = interaction breed by treatment, WHC = water holding capacity, SF = shear force, *L** = lightness, *a** = redness, *b** = yellowness, HUE = HUE angle. a–c lowercases denote significant differences according to the declared *p*.

**Table 3 foods-14-01059-t003:** Color attributes of rabbit meat from different breeds (Azteca Negro, AN or Flemish Giant, FG) under different ultrasonication treatments (means ± E.E.).

Color Descriptor	Azteca Negro		Flemish Giant		*p*		
	Control	HIU	Control	HIU	B	TRAT	B × TRAT
Pale	6.12 ± 0.48	6.29 ± 0.48	5.11 ± 0.48	6.4 ± 0.48	NS	0.0407	NS
Pearly	4.78 ± 0.55	4.45 ± 0.55	4.17 ± 0.55	4.12 ± 0.55	NS	NS	NS
Shiny brown	3.53 ^b^ ± 0.46	3.77 ^b^ ± 0.46	4.93 ^a^ ± 0.46	3.22 ^b^ ± 0.46	NS	NS	0.0054
Pink	3.97 ± 0.52	4.27 ± 0.52	5.76 ± 0.52	6.04 ± 0.52	0.0199	NS	NS
Whitish	3.58 ± 0.57	3.62 ± 0.57	3.2 ± 0.57	3.45 ± 0.57	NS	NS	NS

NS = no significant statistical differences were observed (*p* > 0.05). HIU = treatment high-intensity ultrasound, TRAT = effect of HIU application, B = effect of breed, B × TRAT = interaction breed by treatment. Sensory attributes evaluated on a 10 cm linear scale (0 = less intense and 10 = higher intensity). a, b lowercases denote significant differences according to the declared *p*.

**Table 4 foods-14-01059-t004:** Texture attributes of rabbit meat from different breeds (Azteca Negro, AN or Flemish Giant, FG) under different ultrasonication treatments (means ± E.E.).

Texture Descriptor	Azteca Negro		Flemish Giant		*p*		
	Control	HIU	Control	HIU	B	TRAT	B × TRAT
Elastic	3.21 ± 0.45	3.02 ± 0.45	3.11 ± 0.45	3.209 ± 0.45	NS	NS	NS
Hard	2.88 ± 0.46	2.58 ± 0.46	2.21 ± 0.46	1.38 ± 0.46	NS	NS	NS
Fibrous	3.87 ± 0.66	3.46 ± 0.66	4.49 ± 0.66	4.05 ± 0.66	NS	NS	NS
Juicy texture	5.01 ± 0.42	5.81 ± 0.42	4.48 ± 0.42	4.109 ± 0.42	NS	NS	NS
Soft	6.67 ± 0.28	7.03 ± 0.28	6.81 ± 0.28	7.41 ± 0.28	NS	0.0243	NS

NS = no significant statistical differences were observed (*p* > 0.05). HIU = treatment high-intensity ultrasound, TRAT = effect of application of HIU, B = effect of breed, B × TRAT = interaction breed by treatment. Sensory attributes evaluated on a 10 cm linear scale (0 = less intense and 10 = higher intensity).

## Data Availability

The original contributions presented in this study are included in the article. Further inquiries can be directed to the corresponding authors.

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
