# Peer review of "Physicochemical and Sensorial Profile of Meat from Two Rabbit Breeds with Application of High-Intensity Ultrasound"

_foods, 2025, doi:10.3390/foods14061059_

Round 1

Reviewer 1 Report

Comments and Suggestions for Authors
General comments:
Improving the palatability of rabbit meat has commercial value. However, it was not made clear what is new this research was to address. This manuscript would benefit from some reorganization. For instance, in the discussion section, WBS is discussed then instrumental color, followed by collagen.  A more coherent discussion would entail organizing dependent variables (pH, WHC, WBS, and collagen) to address tenderness effects.  

Detailed comments referenced by line (L) number and tables.

L60 In this section indicate the number of rabbits per breed. Need to indicate the sex of the rabbits.
L25 should it be breed instead of race?
L64 Need to clarify if carcasses were frozen before vacuum packaged.
L65 add … “for” 24 hr … maintained at a …
L68 Add how long carcasses were stored frozen.
L68 Need to describe the condition of the carcass at the time HIU applied. Frozen? Thawed if so how thawed and how long? HUI applied to the vacuum package carcass?
L28 Need to clarify if one anatomical side of each rabbit was HIU treated and the other side served as the control.
L69 Need to clarify meaning of repetitions. Or replications with individual animal per breed serving as the experimental unit.
L75 Reword: The variables which included …
L79 Where on the biceps was the color determined? External muscle or cross section (internal)? Indicate the number of measurements taken on each muscle.
L80 Need to indicate how Minolta calibrated, light source, viewing angle, .  
L85 Indicate the number of measurement taken on each muscle.
L86 Need to indicate longissimus was cooked in a vacuum package or otherwise?
L94 Need to indicate sampling number (duplicate or triplicate?) Indicate what muscle(s).
L101 Indicate how packaged during cooking.
L105 For one session the number of samples would be excessive:  2 breeds x 5 animals x 2 trt x 3 muscles= 60 samples.
L118 Should qualify this sentence because the prior sentence focused on breed, which infers the significant effects indicated were on breed.
L120 HUE not significant at p=0.05. Chroma results not shown in table 2.
L131 Should indicate TRAT effect on pale.
L156 English needs to be corrected
L168 change shoed to showed
L174 Consider a better word choice than hard. Or indicate … hard (less tender)  
L196 Needs a reference.
L208 Drop “of then”
L214 Was temperature measured in the current research?
L224 tone or angle?
L233 Since collagen is determined by the hydroxyproline content, additional discussion on how HIU causes the reduction in hydroxyproline would be of value. Does it in fact breakdown this amino acid?  Discussion should be included on relating the projects collagen results to the WBS results.  
L263 The 19. Seems out of place

Table 2:
Need to add superscripts to means for collagen to know which are different.
Footnote indicates hue tone.  Hue is an angle.  

Table 3:
In table fix the spelling of GIANT
Fix spelling of application
Need to indicate the sensory scale in the footnote

Table 4
Why is a p value listed for hard since >0.05?
Is it texture or textura?
Fix spelling of application
Need to indicate the sensory scale in the footnote

Comments on the Quality of English Language
Grammatical errors and spelling errors need to be addressed.  Specific manuscript lines detailed in the previous comments.

Author Response

Dear reviewer, thank you for your comments. We deeply appreciate all the comments that have helped us to improve our document. We have replied to all your comments and included all your requests on the manuscript.

Reviewer 1

General comments:

Improving the palatability of rabbit meat has commercial value. However, it was not made clear what is new this research was to address. This manuscript would benefit from some reorganization. For instance, in the discussion section, WBS is discussed then instrumental color, followed by collagen.  A more coherent discussion would entail organizing dependent variables (pH, WHC, WBS, and collagen) to address tenderness effects.

R: Thanks for this comment which help to have a more fluid structure in the manuscript.

Now the order of the discussion goes ph, color, WHC, Collagen, WBSF and Texture.   

Detailed comments referenced by line (L) number and tables.

L60 In this section indicate the number of rabbits per breed. Need to indicate the sex of the rabbits.

R: The information has been added. L: 62-63

L25 should it be breed instead of race?

R: The word was corrected. Line 25

L64 Need to clarify if carcasses were frozen before vacuum packaged.

R: They were frozen before the packaging. We have made clear the idea in the manuscript. Lines 71-73

L65 add … “for” 24 hr … maintained at a …

R: The word was added.  Line 72

L68 Add how long carcasses were stored frozen.

R: The information has been added. Line 73

L68 Need to describe the condition of the carcass at the time HIU applied. Frozen? Thawed if so how thawed and how long? HIU applied to the vacuum package carcass?

R: A more clear description has been added to this section. Line 77-78

L28 Need to clarify if one anatomical side of each rabbit was HIU treated and the other side served as the control.

R: This has been clarified in the abstract and the material and methods section. Lines 76-78

L69 Need to clarify meaning of repetitions. Or replications with individual animal per breed serving as the experimental unit.

R: There was a mistake. We have corrected it to replications. Line 76

L75 Reword: The variables which included …

R: Thanks, the line has been reworded. Line 90

L79 Where on the biceps was the color determined?

R= External Bicep femoris muscle and three measurements per muscle. Line 94

L80 Need to indicate how Minolta calibrated, light source, viewing angle.

R: The information was added. Lines 96-98.

L85 Indicate the number of measurement taken on each muscle.

R: Information added. Line 100-102

L86 Need to indicate longissimus was cooked in a vacuum package or otherwise?

R=The information has been added. Line 104

L94 Need to indicate sampling number (duplicate or triplicate?) Indicate what muscle(s).

R= Six cylinders of 1cm diameter were obtained from the longuissimus dorsi per muscle. Information is added in the manuscript. Line 106

L101 Indicate how packaged during cooking.

R= The information has been added. Line 139

L105 For one session the number of samples would be excessive:  2 breeds x 5 animals x 2 trt x 3 muscles= 60 samples.

R=The requested information was added. Lines 143-145 and 148-149

L118 Should qualify this sentence because the prior sentence focused on breed, which infers the significant effects indicated were on breed.

R: The sentence has been corrected. Lines 159-165

L120 HUE not significant at p=0.05. Chroma results not shown in table 2.

R: Hue has been declared as not significant. The results Chroma were added in Table 2.

L131 Should indicate TRAT effect on pale.

R: All the paragraph was reworded to have a clearer description of effect on color attributes. Lines 173-175.

L156 English needs to be corrected

R: English was corrected. Line 182

L168 change shoed to showed

R: Correction was made. Line 275

L174 Consider a better word choice than hard. Or indicate … hard (less tender)

R: The recommendation was taken. We have changed the term. Line 281

L196 Needs a reference.

R: Reference was added.

L208 Drop “of then”

R: Correction was done. Line 215

L214 Was temperature measured in the current research?

R: Temperature was monitored and kept constant inside the ultrasonication system. Nevertheless, we have adjusted the idea according to this. Lines 220-221

L224 tone or angle?

R: We have changed to angle. Line 231

L233 Since collagen is determined by the hydroxyproline content, additional discussion on how HIU causes the reduction in hydroxyproline would be of value. Does it in fact breakdown this amino acid?  Discussion should be included on relating the projects collagen results to the WBS results.

R=The discussion has been added. Lines 320-328

L263 The 19. Seems out of place

R: It has been deleted. 

Table 2:

Need to add superscripts to means for collagen to know which are different.

Footnote indicates hue tone.  Hue is an angle.  

R: Superscripts have been added, and angle has been corrected.

Table 3:

In table fix the spelling of GIANT

Fix spelling of application

Need to indicate the sensory scale in the footnote

R: Spelling has been corrected, and scaled added.

Table 4

Why is a p value listed for hard since >0.05?

Is it texture or textura?

Fix spelling of application

Need to indicate the sensory scale in the footnote

R: Spelling and mistakes have been corrected, and scaled added.

Grammatical errors and spelling errors need to be addressed.  Specific manuscript lines detailed in the previous comments.

We have re-check the grammatical and spelling. thanks for your observations. 

Reviewer 2 Report

Comments and Suggestions for Authors

The manuscript investigated the physicochemical and sensorial profile of meat from two rabbit reeds with application of high-intensity ultrasound. The objectives are clear and the study has some significance. However, there are still some points that need to be revised.

  • Line 22: Change “L*” into “L*”, and “a*” into “a*”. L and a should be italicized in the manuscript, and the same goes for line26, line27, line75, etc..
  • Please add “rabbit breed” as a keyword to enhance the representativeness of the keywords for the research objects.
  • In Table 4, “Juicy textura” is misspelled; it should be “Juicy texture”. Line139, 144: change “aplicattions” into “applications”.
  • In the Methods section, the calibration information of the ultrasonic equipment, such as probe type, power stability, etc., needs to be supplemented. .
  • The reference numbers and their citations in the main text need to be checked. For example, in the discussion on page 5, "[19]" is cited, but there is no corresponding entry in the reference
  • How were the ultrasonic treatment parameters (20 minutes, 50kHz, 200W) determined? It is necessary to cite literature to support their rationality or supplement the experimental basis for parameter selection.
  • Regarding the materials and methods, the sample size is relatively small. It is necessary to explain this limitation in the discussion section or supplement replicate experiments to improve the reliability of the results.
  • "Water holding capacity" is abbreviated as WHC in the main text, but it is not marked when it appears for the first time. Please give the full name when it appears for the first time. Please check it all over the manuscript.

   Please supplement the ethical approval information for animal slaughter.

Author Response

Dear reviewer, thank you for your comments. We deeply appreciate all the comments that have helped us to improve our document. We have replied to all your comments and included all your requests on the manuscript.

Reviewer 2

The manuscript investigated the physicochemical and sensorial profile of meat from two rabbit reeds with application of high-intensity ultrasound. The objectives are clear and the study has some significance. However, there are still some points that need to be revised.

  • Line 22: Change “L*” into “L*”, and “a*” into “a*”. L and a should be italicized in the manuscript, and the same goes for line26, line27, line75, etc..

R: All the color terms have been changed to italics.

  • Please add “rabbit breed” as a keyword to enhance the representativeness of the keywords for the research objects.

R: Rabbit breed has been added as a keyword.

  • In Table 4, “Juicy textura” is misspelled; it should be “Juicy texture”. Line139, 144: change “aplicattions” into “applications”.

R: Spelling in table has been corrected.

  • In the Methods section, the calibration information of the ultrasonic equipment, such as probe type, power stability, etc., needs to be supplemented.

R=The information has been added. No information about the calibration is described, since as far as our knowledge, the HIU system does not require a calibration every time it is actioned. It requires maintaining service every two years.

  • The reference numbers and their citations in the main text need to be checked. For example, in the discussion on page 5, "[19]" is cited, but there is no corresponding entry in the reference

R: References have been checked and adjusted in the whole document.

  • How were the ultrasonic treatment parameters (20 minutes, 50kHz, 200W) determined? It is necessary to cite literature to support their rationality or supplement the experimental basis for parameter selection.

R: Parameters were determined by previous studies of ultrasonication of meat in beef and rabbits meat. The information has been added to the section to support the experimental basis. Lines 85-87

Reyes-Villagrana, R.A.; Huerta-Jimenez, M.; Salas-Carrazco,  J.L.; Carrillo-Lopez,  L.M.; Alarcon-Rojo, A.D.; Sanchez-Vega, R.; Garcia-Galicia. I.A. High-intensity ultrasonication of rabbit carcasses: a first glance into a small-scale model to improve meat quality traits. Ital. J. Anim. Sci. 2020, 19:544–550. https://doi.org/10.1080/1828051X.2020.1763212.

Carrillo-Lopez, L.M.; Robledo, D.; Martínez, V.; Huerta-Jimenez, M.; Titulaer, M.; Alarcon-Rojo, A.D.; Chavez-Martinez, A.; Luna-Rodriguez, L.; Garcia-Flores, L.R. Post-mortem ultrasound and freezing of rabbit meat: Effects on the physicochemical quality and weight loss. Ultrason. Sonochem, 2021 79:1–11. doi:10.1016/j.ultsonch.2021.105766.

Caraveo‐Suarez, R.O.; Garcia‐Galicia, I.A.; Santellano‐Estrada, E.; Carrillo‐Lopez, L.M.; Huerta‐Jimenez, M.; Alarcon‐Rojo, A.D. Integrated multivariate analysis as a tool to evaluate effects of ultrasound on beef quality. Journal of Food Process Engineering, 2023, 46(6), e14112.

  • Regarding the materials and methods, the sample size is relatively small. It is necessary to explain this limitation in the discussion section or supplement replicate experiments to improve the reliability of the results.

R=The information has been added at the beginning of the discussion. Lines 190-196

  • "Water holding capacity" is abbreviated as WHC in the main text, but it is not marked when it appears for the first time. Please give the full name when it appears for the first time. Please check it all over the manuscript.

R: The term was already described in the abstract. Additionally, by your request we have described too in the first time appearing in the introduction. Line 57

Please supplement the ethical approval information for animal slaughter.

R=The information has been added. Lines 64-69

Reviewer 3 Report

Comments and Suggestions for Authors

The objective of this study was to evaluate the effect of HIU application in meat from two rabbit breeds on its physicochemical parameters and sensory profile. Although the author proposes a method for enhancing the physicochemical and sensory perception of the tenderness in the meat of both breeds, the experimental design is too simple. The authors only simply measured some physicochemical indexes, lacking the determination of structural changes caused by ultrasound treatment. The research lacks innovation and in-depth analysis, which limits the manuscript’s contribution to the field. Additionally, the manuscript also shows up a number of other shortcomings. Therefore, I do not recommend accepting the manuscript.

Author Response

Dear reviewer, thank you for your comments. We deeply appreciate all the comments that have helped us to improve our document. 

Reviewer 4 Report

Comments and Suggestions for Authors

The Authors present a concise research of a practically important food processing method.

Remarks and questions:

  • The number of individual carcasses should be mentioned as a crucial factor in the research.
  • How were the panelists trained before the sessions?
  • It is not clear, whether "TRAT" refers to both breeds globally or not. Why didn't the Authors evaluate the effect of treatment separately for each breed (and eventually globally as well)? Indication of significant differences/homogeneous groups would be an assset.
  • Table 2: values and SD should be given at the same accuracy level. Why are statistical data missing for collagen (B, TRAT)? Why do Chroma values missing?
  • Table 3: the same as above. Why are statistical data missing for "shiny brown" (B, TRAT)?
  • Table 4: please, check accuracy levels!

Author Response

Dear reviewer, thank you for your comments. We deeply appreciate all the comments that have helped us to improve our document. We have replied to all your comments and included all your requests on the manuscript.

Reviewer 4

The Authors present a concise research of a practically important food processing method.

Remarks and questions:

  • The number of individual carcasses should be mentioned as a crucial factor in the research.

R: Number of individuals have been included in abstract and the material and methods section. Lines 62-62

  • How were the panelists trained before the sessions?

R=The information was added. Lines 119-135

  • It is not clear, whether "TRAT" refers to both breeds globally or not. Why didn't the Authors evaluate the effect of treatment separately for each breed (and eventually globally as well)?

R:TRAT refers exclusively to the HIU application. We have corrected in the text any missuse of the word treatment or TRAT to avoid confusion.

The statistical factorial design is the most appropriate for this kind of studies where two factors can interact. If we evaluated separately, statistical information may be loss (i.e. liberty grades, significance, etc) and the interaction cannot be detected.  

  • Indication of significant differences/homogeneous groups would be an assset.

R: If the reviewer refers to some missing superscripts in the tables, we have added them to leave clearer the differences among treatments when necessary.

  • Table 2: values and SD should be given at the same accuracy level. Why are statistical data missing for collagen (B, TRAT)? Why do Chroma values missing?

R: Values and SD have been corrected at the same accuracy level. The values Chroma were included in the table. By mistake, every time the interaction was significant we avoided to present the significance of the single factor, to point out the importance of the interaction. However we have included now the correspondence information in the tables.  

Table 3: the same as above. Why are statistical data missing for "shiny brown" (B, TRAT)?

R=This has been corrected in the table.

  • Table 4: please, check accuracy levels!

R: Accuracy levels have been checked.

Round 2

Reviewer 3 Report

Comments and Suggestions for Authors

This manuscript has been improved. It is, however, strongly recommended to unify the format of references.

Author Response

Thank you for your comments. The references have been updated, organized and re-checked with a reference manager. 

Reviewer 4 Report

Comments and Suggestions for Authors

The questions and remarks were addressed.

Author Response

Thank you for your contribution. We are glad to see your reply.